# Investigating the Role of GABA in Neural Development and Disease Using Mice Lacking GAD67 or VGAT Genes

**DOI:** 10.3390/ijms23147965

**Published:** 2022-07-19

**Authors:** Erika Bolneo, Pak Yan S. Chau, Peter G. Noakes, Mark C. Bellingham

**Affiliations:** 1School of Biomedical Sciences, Faculty of Medicine, The University of Queensland, Brisbane 4072, Australia; e.bolneo@uq.net.au (E.B.); p.chau@uq.net.au (P.Y.S.C.); mark.bellingham@uq.edu.au (M.C.B.); 2Queensland Brain Institute, The University of Queensland, Brisbane 4072, Australia

**Keywords:** GAD67, GAD65, VGAT, GABA, GABA-receptors, GABAergic transmission, glutamatergic transmission, neural development

## Abstract

Normal development and function of the central nervous system involves a balance between excitatory and inhibitory neurotransmission. Activity of both excitatory and inhibitory neurons is modulated by inhibitory signalling of the GABAergic and glycinergic systems. Mechanisms that regulate formation, maturation, refinement, and maintenance of inhibitory synapses are established in early life. Deviations from ideal excitatory and inhibitory balance, such as down-regulated inhibition, are linked with many neurological diseases, including epilepsy, schizophrenia, anxiety, and autism spectrum disorders. In the mammalian forebrain, GABA is the primary inhibitory neurotransmitter, binding to GABA receptors, opening chloride channels and hyperpolarizing the cell. We review the involvement of down-regulated inhibitory signalling in neurological disorders, possible mechanisms for disease progression, and targets for therapeutic intervention. We conclude that transgenic models of disrupted inhibitory signalling—in GAD67^+/−^ and VGAT^−/−^ mice—are useful for investigating the effects of down-regulated inhibitory signalling in a range of neurological diseases.

## 1. Preface

To grasp the impact of down-regulated inhibitory signalling on motor function, cognitive function, and behaviour of the organism at large, we must understand the role of GABA in the nervous system. After exploring the importance of the GABAergic system in normal neurodevelopment, we gain an appreciation for how vital it is to maintain an excitatory–inhibitory signalling balance and how aberrant signalling leads to the pathophysiology of synaptic inhibition disorders. We then outline several disorders of synaptic inhibition including epilepsy, autism spectrum disorders, schizophrenia, and anxiety. Finally, two rodent models of down-regulated GABAergic inhibition are examined—the GAD67^+/−^ [1] and VGAT^−/−^ mouse [2]. These mouse models provide a key tool for investigating the mechanisms that underpin synaptic inhibition disorders. A greater understanding of the impact of dysregulated synaptic inhibition may also lead to a positive change in the field of mental health.

## 2. Importance of GABA in the Nervous System

### 2.1. Inhibition in the Forebrain Is Mediated by GABA

Neural excitation by glutamatergic neurons is the primary influence driving central neurons to fire and is constantly counterbalanced by inhibitory synaptic inputs [3]. Inhibition in the central nervous system (CNS) is elicited by two major neurotransmitters: gamma-aminobutyric acid (GABA), and glycine [4]. GABA is the primary inhibitory synaptic neurotransmitter of the CNS, with GABAergic synapses found throughout the brain [5] and especially in the forebrain, where they predominate. In addition, since many events of neurogenesis occur before the onset of synapse formation, non-synaptic GABAergic transmission together with endogenously released GABA has been postulated to be involved in early neurodevelopment [6]. Tonic GABA release [7,8] lays down the foundation for normal proliferation of neuronal precursors, neuronal differentiation and migration, and early activity patterns [6]. Clearly, proper formation of neuronal networks relies on the GABAergic system to function optimally in the developing brain.

The distribution of GABAergic circuits has been defined using immunohistochemical and electrophysiological techniques [4]. GABA is synthesised by GABAergic interneurons, and elicits inhibition by binding to GABA receptors on the postsynaptic membrane of other neurons [9]. GABAergic circuits are widely distributed in CNS regions such as the cortex, hippocampus, thalamus, hypothalamus, brainstem, and basal ganglia [5,9,10,11,12]. It is this sculpting of excitatory transmission by the inhibitory GABAergic system that allows normal synapse formation, maturation, and maintenance of neural circuits within the CNS [4,13]. GABAergic transmission is required for modulating circuits involved in complex cognitive behaviours, including personality expression, decision making, and goal-orientation [14].

Several brain regions use both GABAergic and glycinergic inhibition, including the retina, spinal cord, cerebellum, brainstem nuclei, olfactory bulb, and hippocampus [15,16,17,18,19,20]. Within these regions, GABAergic and glycinergic inhibition can act independently or together to modulate excitatory signals [21]. Mixed inhibitory signalling is also evidenced by the co-release of GABA and glycine from the axon terminal of brainstem and spinal cord interneurons, allowing a wider dynamic range of inhibitory control [22,23].

### 2.2. The Role of GABA in the Mature Central Nervous System

The effects of GABA are elicited by the binding of GABA to the ionotropic GABA-A and GABA-C receptor subtypes and the metabotropic GABA-B receptor subtype on the postsynaptic membrane of neurons [24,25] (Figure 1). This review will focus on the role of GABA-A receptors, which form the majority of GABA receptors within the CNS. GABA-A receptors incorporate an ion channel that allows the passage of chloride anions, and exert primary control over inhibitory signalling [26]. Not only do GABA-A receptors predominantly control inhibitory signalling within the CNS, they also play a vital modulatory role in neurodevelopmental events leading to the establishment of complex neuronal networks and to behaviours they regulate [26,27]. GABA-A receptors also exhibit binding sites for several modulatory molecules, including ethanol, benzodiazepines, anaesthetics, neurosteroids, barbiturates, and picrotoxin [26]. After GABA binding to their receptors, GABA is removed from the synaptic cleft into glial cells via GAT 2/3 for breakdown into glutamine, or transported into the presynaptic terminal via GAT-1 for recycling into synaptic vesicles, the latter accounting for ~80% of GABA uptake ([28] Figure 1). Additionally, Neuroligin-2 (NL2) is a postsynaptic cell adhesion protein exclusively localised at GABAergic synapses [29] that mediate a bidirectional signalling between pre- and postsynaptic neurons by forming a trans-synaptic signal transduction complex [30]. In the process of forming the complex, NL2 is accountable for recruiting required additional proteins, which is essential for the stabilisation/destabilisation of GABAergic synapses [31,32].

Gamma aminobutyric acid (GABA) is synthesised by glutamic acid decarboxylase (GAD) [33] (Figure 1). GAD comes in two isoforms—67 kDa (GAD67) and 65 kDa (GAD65), which are derived from different genes, rather than alternatively spliced proteins that come from the same gene [34]. GAD67- and GAD65-mediated synthesis of GABA differ in a temporospatial manner [35]. GAD67 contributes to over 90% of basal GABA synthesis and is distributed throughout the cell, while GAD65 remains localised to presynaptic nerve terminals [35] (Figure 1). During development, synthesis of GABA by the two enzymes GAD65 and GAD67 also changes significantly [36]. GAD67 deficient (KO; GAD67^−/−^) mouse brains contain 70–95% less GABA at birth, compared to wild type (WT; GAD67^+/+^) mice, and die at birth due to respiratory failure [36]. By contrast, the brains of GAD67^+/−^ mice are 30–50% deficient in GABA at birth and pups survive to adulthood [36]. GAD65^−/−^ mice have normal levels of GABA at birth but show a progressive decrease in GABA levels in the hippocampus and cortex from 60 days postnatal [37]. While both heterozygous (+/−) GAD67 and GAD65 mice exhibit spontaneous seizures and behavioural deficiencies, only the latter are predisposed to premature death [37,38]. GAD67 activity thus provides most GABA synthesis during prenatal and early postnatal (P) development to day 60 (P60), while GAD65 is the predominant source of GABA in the adult CNS [36].

### 2.3. The Role of VGAT in GABA and Glycine Signalling

The vesicular GABA transporter (VGAT), also known as vesicular inhibitory amino acid transporter (VIAAT), is a common vesicle transporter for GABA and glycine, and is essential for normal GABAergic and glycinergic neurotransmission [39,40]. It facilitates GABA and glycine signalling by transporting them into their synaptic vesicles prior to exocytosis [41] (Figure 1). The uptake of GABA and glycine by VGAT is heavily dependent on pH and electrochemical gradients across the vesicular membrane that are regulated by Mg^2+^-activated ATPase [41]. VGAT is highly expressed in the nerve terminals of both glycinergic and GABAergic neurons [39,40]. In regions where GABA and glycine are co-released, GABA has a higher affinity towards VGAT than glycine [41].

### 2.4. GABA in Neurodevelopment

A growing focus of research is dedicated towards revealing the critical role of inhibitory neurotransmitters in refining many aspects of neurodevelopment [7,8,42]. In addition to inhibitory signalling in the mature CNS, GABA has been demonstrated to provide significant excitatory activity in the developing CNS [43,44]. Immature neurons use ionotropic GABA transmission by transporting chloride (Cl^−^) anions across the cell membrane, where elevated intracellular Cl^−^ concentration causes the cell membrane to depolarise and elicit functionally excitatory actions [45,46]. During two weeks postnatal in rodents and approximately full-term (forty weeks) birth in humans, the efflux of Cl^−^ from immature CNS neurons is increased, causing the neurons to shift from depolarising to hyperpolarising [46]. This electrochemical shift is necessary for establishing the subsequent inhibitory action of GABAergic neurons, playing a vital role in the developing CNS. Evidence suggests a combination of excitation from GABAergic, glycinergic, and glutamatergic neurons contribute to activity-dependent remodelling during neurodevelopment [13]. Each of these neurotransmitter systems work together to delicately balance neural activity within a normal physiological range, thus playing a key role in establishing neural circuits of neural pathways necessary for vision, hearing, breathing, and pain [42,47].

GABAergic synaptic inputs mould connectivity and plasticity of the developing mammalian brain [48]. While GABA is the major inhibitory neurotransmitter in the mature mammalian CNS, GABA-A receptors evoke excitation before birth and this period of excitation can even extend into early postnatal life, evoking depolarisation in postsynaptic neurons [46,49,50,51,52,53,54]. Through this signalling, postsynaptic neurons, which are new projection neurons derived from neurogenesis of proliferative zones such as ventricular and subventricular zones [55], are guided to migrate into target brain areas [7,48,56]. This process of migration is dependent on the receptor subtype expressed on the postsynaptic neuron [50]. Neural progenitors in the proliferative zone express GABA-A receptors [57,58,59]. Here, GABAergic signalling can promote proliferation, cell migration, and cell cycle exit [59]. Once cells reach the cortical plate, GABA-A receptor activation results in migration cessation [59,60]. Once foundational GABAergic circuits are in place, continued excitatory signalling through GABA-A activation causes projection neurons to extend dendrite length and form new synaptic contacts [61]. Along with promoting normal development of neural processes, GABA-A receptor activity also regulates the maturation of inhibitory synapses required for differentiation [50,62]. Dynamic expression of GABA receptor subtypes and their respective activity constitute a careful orchestration directed by the GABAergic system in normal CNS development [50].

The shift from an excitatory to an inhibitory action of GABA may also be modulated in an activity-dependent manner [49]. Following excitatory GABAergic activity and near the time of birth, increased expression of the potassium chloride transporter 2 (KCC2) switches the transmembrane chloride gradient from a depolarising to a hyperpolarising direction, thereby changing GABAergic signalling to inhibitory [50]. Through hyperpolarisation, the postsynaptic neuron’s ability to fire action potentials is down regulated [63], allowing for fine tuning of neural information processing [64].

In light of this, genetic mutation of any of the components of the GABAergic system have been shown to cause a wide range of neurological disorders [3]. Abnormal development of the GABAergic system could result either in disorganised neural circuits caused by improper GABA function during development, and/or by alteration in GABAergic synaptic function resulting in atypical action potential firing of neural circuits, underlying the pathophysiology of many synaptic inhibition disorders [50].

### 2.5. Excitatory–Inhibitory Balance Is Crucial in Normal Neurodevelopment

Excitatory–inhibitory (E-I) balance in neuronal circuits has become of increasing research interest, in most part due to its potential roles in the aetiology of a wide range of neurological diseases [3]. Neuronal circuits coordinate E-I signalling that varies depending on the individual’s developmental period and interactions with the environment [13,65]. During development, the quantity and distribution of excitatory versus inhibitory synapses across neurons and their individual dendrites serve as the hardware of the brain [13,66]. These cellular computations allow us to perceive our environment and interact accordingly. There is a strong relationship between altered synaptic activity onto neurons (as measured by patch-clamp electrophysiology) and concomitant neuronal dendritic architecture and spine density changes (as measured by single neuron morphology) [13,67,68,69]. This association between abnormal synaptic physiology and dendritic structures occurs in both pyramidal cells and other neuronal types [13,67]. These relationships between alterations in neuronal structure and function have been confirmed using other techniques, such as Golgi impregnation [67,70].

Strong evidence suggests an increased ratio of excitatory over inhibitory (E/I) transmission contributes to the pathogenesis of neurological disorders, with a large number of mutations and variants of different genes culminating to a disturbed E-I balance [50]. Mutations in genes leading to dysfunctional inhibitory synapses and inhibitory neurotransmission thus place an individual at higher risk of developing neurological disorders [50,71]. A substantial proportion of clinically identified neurodevelopmental disorder risk genes encode GABAergic transcription factors, GABA receptors, chloride transporters, and inhibitory synaptic proteins [48]. An increased E-I ratio has been found in animal models for epilepsy, schizophrenia, ASDs, and anxiety [35,71]. Additionally, mutations in genes coding for the enzymes responsible for glycine synthesis have been seen to cause hyperglycinemia—a disease characterized by mental retardation, lethargy, and seizures [72].

More importantly, in the instance of inhibitory signalling, an altered GABAergic system, including GABAergic interneuronal loss, disrupted cell maturation, imbalance in GABA-mediated inhibitory synaptic transmission, or reduced GABA, GAD, parvalbumin (PV), and somatostatin (Sst) expression, have been reported in a variety of neurodevelopmental and neuropsychiatric disorders [73,74,75,76,77].

To understand the vital role GABAergic signalling plays in establishing normal neurodevelopment, and how mutations within genes of the GABAergic system can have devastating impacts in development and the organism at whole, we will examine two rodent models for down-regulated GABAergic influence—the GAD67^+/−^ and VGAT^−/−^ mouse. The GAD67^+/−^ mouse allows us to investigate the effects of a 50% global reduction of GABA in the developing CNS and how a mutation within the gene that codes for GAD67, can manifest in neurological disease [36,78]. Another mouse model worthy of importance when examining GABAergic signalling in normal neurodevelopment and neurological diseases is the vesicular GABA transporter (VGAT) deficient mouse model (i.e., the VGAT^−/−^ mouse). This review assesses the use of these two mouse models when investigating potential therapeutics for epilepsy, schizophrenia, ASDs, and anxiety.

## 3. Disorders of Synaptic Inhibition

### 3.1. Synaptic Inhibition Dysfunction in Epilepsy

Epilepsy is a common heterogeneous neurological symptom affecting 70 million people globally [79,80]. It is most prevalent in infants (<1 year) and at older ages (>50 years) [80]. It is characterised by spontaneous recurrent neuronal hyperexcitability, known as seizures, which are observed in a range of diseases with different underlying aetiologies [81,82]. Seizures are divided into generalised seizures and focal seizures. General seizures affect both cerebral hemispheres, whereas focal seizures typically only affect neuronal circuits in one hemisphere [83]. The behavioural outcome and the severity of a seizure critically depend on the brain region where hyperexcitability occurs. Emerging evidence suggests that epilepsy has a strong genetic predisposition in most cases. Up to 50% of epilepsy patients have comorbidities, such as psychiatric conditions or somatic disorders [84]. These observations help to advance our understanding of the underlying cause and pathophysiology for individual patients. Antiseizure medication can often help two-thirds of epileptic patients to achieve long-term remission from seizures; however, up to a third of patients have drug-resistant epilepsy which might require surgery. Overall, a curative treatment for epilepsy is not yet available and its premature mortality rate poses a pressing public health problem.

Extensive research has been done on how seizures are generated and propagate in the brain. However, the precise mechanism of epileptogenesis remains unclear, due to the wide variation in aetiology of epileptogenesis. The causes and risk factors of epileptic seizures are usually categorised into six groups: genetic, structural, metabolic, infectious, immune, and idiopathic [83]. One of the prominent epileptogenesis mechanisms is a result of an episodic shift in the balance of excitatory/inhibitory (E/I) synaptic transmission towards excitation [85]. Many animal studies and human observational studies have shown that impaired synaptic inhibition plays a significant role in epileptogenesis [77,86]. The hippocampus and cerebral cortices are most prone to epileptogenesis, as they exhibit the lowest seizure thresholds [87]. Since the GABAergic interneuron system is crucial for balancing E/I synaptic transmission in these regions, reduction in inhibitory input would promote glutamatergic hyperexcitability commonly seen in seizure generation. In fact, an impaired GABAergic interneuron system is often seen in temporal lobe epilepsy (TLE). TLE patients who were exposed to brain insult such as brain trauma or meningitis in early childhood tend to develop seizures later in life. A drastic decrease in parvalbumin (PV)-positive interneurons is also seen in epileptic human dentate gyrus [77]. Up to 70% of epilepsy patients develop hippocampal sclerosis [88]. For instance, TLE patients are found with neuronal loss, including GABAergic interneurons, in hippocampal CA1/CA3 regions and amygdala [89,90,91,92].

Animal studies have induced epilepsy through many approaches, including genetic, pharmacological, and physical interventions, designed to model the molecular and pathophysiological events in human epilepsy as closely as possible [93,94]. Several gene mutations in pre- and post-synaptic proteins have been discovered in idiopathic epilepsy. One example is that of SCN1A, one of the genes responsible for generalized epilepsy with febrile seizures plus (GEFS+). SCN1A encodes the α-subunit of the neuronal sodium channel NaV1.1, which is concentrated in the axon initial segment of parvalbumin positive GABA interneurons [95,96]. Mutations to SCN1A reduces the excitability of these GABAergic inhibitory interneurons, resulting in hyperexcitation in the thalamus, cortex, and hippocampus in SCN1A mutant mouse models carrying the same NaV1.1 mutation found in humans [96,97,98,99].

Hyperexcitation in normal brain tissues can also be achieved by pharmacologically downregulating neurotransmitter- or voltage-gated ion channels inhibiting neuronal activity, such as GABA-A receptors or potassium channels [100,101], or by enhancing neurotransmitter- or voltage-gated ion channels that excite neuronal activity, such as glutamate receptors or sodium channels [102]. For instance, blocking GABA_A_ receptor-mediated inhibition by administering GABA_A_ receptor antagonists (bicuculline, picrotoxin or penicillin) could induce epileptiform discharges in cortical regions both *in vivo* [100,103,104] and *in vitro* [105,106,107]. Contrarily, these epileptic effects can be reversed by increasing synaptic inhibition [108] or decreasing excitation [109]. Focal cortical dysplasia (FCD) is a physical condition with abnormal brain cell development and organisation that is a common and important cause of medically intractable epilepsy [110]. A significant reduction in GABAergic interneuron numbers is reported in both human and rodent brain in FCD [110,111]. Reduction in IPSC frequency, cumulative GABA-mediated synaptic currents, and abnormal GABA reuptake is also seen in FCD rodent models [111], demonstrating that compromised synaptic inhibition plays an essential role in the process of seizure genesis. In summary, many studies have highlighted the contribution of impaired synaptic inhibition on the pathophysiology and generation of epilepsy.

### 3.2. Synaptic Inhibition Dysfunction in Schizophrenia

Schizophrenia (SCZ) is a complex, debilitating, chronic neuropsychiatric disorder characterised by a range of symptoms including auditory hallucinations, delusions, social functioning deficits, and cognitive impairments [112,113,114]. The development of SCZ is suspected to be an interplay between environmental risk factors and genetic predisposition [115,116,117]. It is a highly heritable disorder—studies have shown that first and second-degree relatives are approximately 10% and 3% at risk of SCZ respectively, while the risk of having SCZ when both parents are schizophrenic is up to 40% [118]. An individual could be more vulnerable to developing SCZ with a combination of common environmental and social factors, including childhood trauma, social isolation, and discrimination [114]. Given that fewer than 14% of patients can sustain a recovery after their first psychotic episode, it is crucial to uncover the pathogenesis of SCZ.

However, the heterogeneous aetiology of SCZ has posed a challenge in understanding the molecular and cellular mechanisms in its pathogenesis. Existing/established theories are centred around an imbalance of different neurotransmitters, such as dopamine, serotonin, glutamate, aspartate, glycine, and GABA [114]. Multiple lines of evidence suggested that a defect in inhibitory GABAergic neural circuits plays an important role in the pathophysiology of SCZ. Alternation of different components in the GABAergic system is a consistent finding in post mortem schizophrenic brain issues of both human and animal models. Significant reductions of GAD67 mRNA and GAD67 protein expression are observed in different regions of the cortex, including the dorsolateral prefrontal cortex (DLPFC) [119,120], anterior cingulate cortex (ACC) [121], and orbital frontal cortex (OFC) [74], as well as the cerebellar cortex [74,122]. More importantly, decreased GAD67 mRNA expression appears to be concentrated in a subset of parvalbumin (PV)-expressing GABAergic interneurons [119,123], as GAD67 expression is unaltered in inhibitory chandelier neurons [124]. In addition to the brain areas mentioned above, the primary visual cortex (VC), primary motor cortex, and superior temporal gyrus also exhibit similar alteration in GABAergic gene expression, including reduction in GAD67 mRNA, GAD65 mRNA, GAT-1 mRNA, and GABA_A_ receptor α1 and δ subunits mRNA [74,125,126]. The GAT-1 is a transporter found abundantly in GABAergic interneurons which is responsible for removing GABA from the synaptic cleft and curtailing GABAergic neurotransmission [127,128]. Unlike VGAT, which is highly concentrated in nerve terminals, GAT-1 is expressed in both synaptic terminals and axons [41]. Hence, reduction of GAT-1 could lead to prolonged accumulation of GABA at the synaptic cleft and increased GABA-mediated inhibitory synaptic transmission [129,130,131]. However, since there is a general downregulation of both GAD67 mRNA and GAD67 mRNA, the effect of decreased GAT-1 could be a compensatory phenomenon to maintain a certain level of GABAergic transmission. Abnormal GABAergic expression in these brain regions is thought to underlie some of the schizophrenic symptoms, including auditory hallucinations [74] and deficits in social function [132]. However, these post mortem studies cannot confirm whether impairments in the GABAergic system, primarily in PV interneurons, is the cause or an effect of SCZ.

A wide range (>20) of animal models of SCZ have been developed to investigate the many pathways of SCZ pathogenesis through four induction categories: developmental, pharmacological, lesion, and genetic manipulations [133]. The main hypothesis used when developing rodent models has evolved from hyper-dopaminergic (dopamine centred) [134,135] and hypofunction of NMDA receptor (glutamatergic centred) [136,137] to GABAergic centred more recently [73,138]. In addition, the majority of these models were able to mimic at least part of the SCZ phenotype, including reduced social interaction and increased aggression.

As GABAergic system dysfunction is a common finding in SCZ patients, a handful of animal models with GABAergic alternation were generated to investigate the relationship between SCZ-like behaviours and post mortem brain tissue findings. Several behavioural phenotypes in rodents are thought to be relevant to schizophrenic symptoms in patients, including locomotor hyperactivity, abnormal social behaviours, sensorimotor gating deficits, and cognitive impairment [139]. GAT-1 deficient mice exhibit schizophrenia-like behavioural abnormalities, such as significantly higher locomotor activity, impaired object recognition memory, impaired sensorimotor gating, and abnormal social behaviour [140]. Reduction in GABA reuptake in these mice results in an increase in tonic GABA currents in the prefrontal cortex [140]. Common antipsychotics were able to reverse some of the schizophrenic-like behaviours in GAT-1 deficient mice [140]. These results show the potential role that GAT-1 and GABA abnormalities play in the pathogenesis of SCZ.

Furthermore_,_ GAD67 heterozygous (GAD67 ^+/−^) mutant mice show an overall 40% reduction in GAD67 expression in the brain and 16% reduction in GABA levels [141]. GAD67^+/−^ mice specifically lacking GAD67 expression in PV interneurons are found to display schizophrenia-like behaviour such as deficits in prepulse inhibition, MK-801 sensitivity, and social memory [73]. In addition, reduced inhibitory synaptic transmission, alternation in NMDA receptor-mediated synaptic responses in pyramidal neurons, and increased spine density in hippocampal CA1 apical dendrites were reported [73]. These results implicate that down-regulated GAD67 in PV interneurons and abnormal glutamatergic excitatory synaptic activity could underlie SCZ pathogenesis. Lastly, a transgenic mouse line with neuroligin-2 (NL2) R215H mutation and a heterozygous neuregulin 1 (NRG1^+/−^) mutation rodent model also presented SCZ-like behaviour abnormalities along with impaired GABAergic transmission [142,143,144].

### 3.3. Autism Spectrum Disorders and GABAergic Dysfunction

Autism spectrum disorder (ASD) and associated neurological disorders have been linked to mutations in genes affecting the ratio between excitatory/inhibitory (E/I) neurotransmission [145]. The impacts of these mutations on neural circuitry are complex and so the aetiology of ASD remains relatively ambiguous [145]. ASD is a combination of neurobehavioural symptoms, defined by the Diagnostic and Statistical Manual of Mental Disorders (DSM-IV) as having restricted interests, poor sociability, and communication deficits [146]. The global prevalence of ASD continues to rise, with approximately 1 in 68 children currently identified as having ASD [147]. The onset of these symptoms occurs during early childhood development [146,147,148] and can thus be diagnosed before the age of 3 years [50]. Previous studies have found abnormal social interaction and blood biochemistry as early as a few days postnatal [33], giving rise to the possibility that early diagnosis and thus early interventions may prevent the pathogenesis of ASD [4,149].

ASD is highly heritable and sexually dimorphic, with males constituting ~80% of overall cases [150]. Early work by Rubenstein and Merzenich (2003) theorised that an increased E/I ratio led to hyperexcitability of cortical circuits in ASD patients and emerging evidence from human and animal models have added to the growing body of work supporting the potential role of abnormal GABAergic neurotransmission in ASD [35]. Dysfunctional GABAergic signalling leading to a disrupted E/I circuit balance during critical periods of neurodevelopment may thereby provide a unifying explanation for ASD development [35,151].

A genetic cause for ASD suggests that aberrant brain development first occurs during embryogenesis [71]. The hypothesis that GAD67 plays a role in the pathogenesis of ASD is due to the fact that down-regulated inhibitory signalling has been found to affect neural networks during critical periods of development [151]. During the early stages of brain development, neuronal circuits are highly plastic and sensitive to slight imbalances in E/I signalling [151]. It is during this stage that single neurons develop multiple functions that are acquired in an experience-dependent manner [151]. Earlier research in the field of developmental neuroscience proposed a consistent and tightly timed schedule for the developing brain [152]. According to the postulation, the stages of neurodevelopment follow: (1) proliferation of undifferentiated brain cells; (2) cell migration towards predetermined brain regions and commencement of cellular differentiation; (3) grouping of similar cell types into distinct brain regions; (4) formation of connections between neurons (synapses) including both short and long-range projections; and (5) competition among connections, resulting in the selective elimination and refining of synapses [152]. In addition to the fundamental processes that form the functioning brain, the natural phenomena known as ‘neuroplasticity’ facilitates further pruning and refining of neural connections, synapses, and dendritic arborisations [153,154]. These changes are stimulated in response to interaction with the environment [154]. Neuroplasticity, therefore, should be better described as experience-driven brain plasticity [154]. Taken together, minor deviations from normal neurodevelopment resulting in aberrant transmission have been linked to abnormal brain development and ASD [71].

Additionally, ASD core behavioural symptoms are associated with the comorbid neurological disorders discussed above, such as epilepsy and schizophrenia, as well as other conditions associated with dysfunctional GABAergic neurotransmission, such as Parkinson’s disease (PD), bipolar disorder, anxiety, and depression [35,71,155].

## 4. GABA Deficient Mouse Models

### 4.1. The GAD67^+/−^ Mouse

The importance of homeostasis involving the GABAergic system during pre- and post-natal development has been highlighted previously. Disrupted GABAergic neurotransmission is found in a variety of neurological conditions, including schizophrenia, ASD, epilepsy, and Alzheimer’s disease (AD). The significance of GAD67 for GABAergic interneuron development and GABA production is well established by multiple studies, including clinical, genetic mutation, and pharmacological manipulation (*reviewed above*).

Tamamaki and colleagues created GAD67 haplodeficient (GAD67^+/−^) mice by replacing one of the GAD67 alleles with a green fluorescent protein (GFP) gene, where both genes were connected to the same promoter enhancer and suppressor in the introns or in the 5′ and 3′ flanking region in the GAD67 allele [141]. The colocalization of GFP with GAD67 and GABA was confirmed via immunohistochemistry. They found that all GFP-positive cells were GAD67 and GABA positive. Although GAD67^+/−^ mice mainly display normal growth, reproductive behaviour, and brain morphology at a macroscopic level, a significant overall reduction in GABA content was observed in juvenile mice. Multiple research groups have since investigated the effect of GAD67 haplodeficiency. Up to 40% and 16% reduction were seen respectively in GAD67 expression and GABA level in the brains of GAD67^+/−^ mice [36,141,156]. Several studies have confirmed that GAD67^+/−^ mice do exhibit a significant decrease in GAD67 expression and that most of the GFP-positive cells accurately reflect GABA-immunopositive neurons [157,158,159]. Since PV-positive cells account for the majority of GABAergic interneurons and GFP-positive cells were immunopositive for PV, it was evident that expression of GAD67 in GABAergic interneurons were reduced by approximately 50% in GAD67^+/−^ mice [157]. Moreover, the GAD67^+/−^ mouse has allowed us to further uncover anatomical features of GABAergic neuronal network, the development of GABAergic interneurons, and their electrophysiology [160,161,162,163]. GABAergic transmission and developmental elimination of climbing fibre to Purkinje cell synapses were reduced in GAD67^+/−^ mice and synapse elimination could be rescued by benzodiazepine treatment [164]. By reducing GAD67 expression conditionally in basket cells, deficits in axon branching and abnormal pyramidal neuron synaptic formation were also observed in adult mouse visual cortex [165]. Disrupted GABAergic inhibitory circuits resulted from reduced axon branching, number of GABAergic interneurons, and impaired cell maturation, due to the chronic reduction in both GAD67 content and GABA level. Supporting these morphological defects in GABAergic neurons are electrophysiological studies using brain slices from GAD67^+/−^ mice, which revealed substantial deficits in inhibitory synaptic transmission, linked to the disinhibition of pyramidal neuron spiking and increased excitation/inhibition balance in PV interneurons [163]. Despite a compensatory role of GAD65 in pre- and post-natal development, a 50% drop in GAD67 (GAD67^+/−^ mice), results in impaired GABAergic development with persistent deficits in synaptic transmission and network dysfunction well into adulthood.

In addition, altered social behaviour is one of the most consistent findings of behavioural studies in GAD67^+/−^ mice. Sandhu and co-workers revealed a reduction in sociability and lowered intermale aggression in GAD67^+/−^ mice [38]. In addition, GAD67^+/−^ mice also displayed impaired sensitivity towards social and non-social odours, suggesting that the neural process of detecting socially related olfactory stimuli was disrupted in GAD67^+/−^ mice [166]. Since both social aggressive and interaction in mice are largely dependent on pheromones, GABAergic neurotransmission has therefore been implicated in the neural processing of olfactory stimuli that control aggression [167,168]. Interestingly, disturbance of olfactory function and its impact on social drive is also observed in schizophrenia patients [169]. Apart from impaired sociability, increased vulnerability to social stress, depressive-like behaviour, and altered catecholaminergic innervation in brain areas linked to schizophrenia are also reported in GAD67^+/−^ mice [170]. Depressive-like behaviour often associated with lowered social drive, is one of the negative schizophrenic symptoms [171]. Additionally, Smith and colleagues reported mild locomotor hyperactivity and altered anxiety-like behaviour in GAD67^+/−^ mice [172], which could be a result of imbalanced E/I due to impaired cortical GABAergic transmission, resembling attention-deficit/hyperactivity disorder (ADHD) that often presents with Tourette syndrome [173,174,175]. The phenotype of GAD67^+/−^ mice in these studies sheds light on the critical role of GAD67 and GABA in the development of social behaviour and motor activities, which share similarity with both behavioural and morphological characteristics for several neurodevelopmental disorder such as schizophrenia, ADHD, and ASD.

Meanwhile, maternal stress (MS) and GAD67/GABA deficiency are both risk factors for developing many neurodevelopmental disorders, including schizophrenia, ASD, ADHD, Tourette syndrome, and anxiety [176,177,178,179]. For example, Uchida and co-researchers have demonstrated that GAD67^+/−^ mice are more susceptible to MS [180,181]. The interaction of both GAD67 haplodeficiency and MS could specifically disrupt PV-positive GABAergic interneuron proliferation, as MS in the GAD67^+/−^ foetus reduced GFP-positive GABAergic interneurons pre- and postnatally in various brain regions [181]. Thus, GAD67^+/−^ mice could provide a useful model for investigating the impact of lowered GABA levels on social behaviour and its interaction with environmental risk factors that could lead to neurodevelopmental disorders.

Lastly, existing literature has revealed that there is an abnormal increase in astrocytic GABA in both ASD mouse models and post-mortem human Alzheimer’s disease (AD) brain tissue [182,183]. Consequently, Wang and colleagues investigated the impact of GABA in an AD mouse model by crossing GAD67^+/−^ mice with 5X FAD mice that express five familial AD gene mutations [157]. In these mice, they found that astrocytic GABA was reduced, resulting in lower levels of amyloid β plaques, a hallmark of AD [184], and importantly neuronal loss around Aβ plaques commonly seen in AD pathology was also reduced [157]. In addition, they also observed reduced tonic GABA inhibition, when compared to controls. Collectively, these findings widened our understanding of the role GABA plays in AD pathology, and show that GAD67 haplodeficiency in AD mouse brains can be beneficial to both pathological and behavioural outcomes in AD, providing a potential avenue for AD therapeutic intervention.

### 4.2. The VGAT Deficient Mouse

GABA and glycine are both transported into synaptic vesicles by a single transporter protein, vesicular GABA transporter (VGAT), also known as vesicular inhibitory amino acid transporter (VIAAT). Uptake of these neurotransmitters into synaptic vesicles is necessary before their exocytosis into the synaptic cleft by both GABAergic and glycinergic neurons [2,40,185]. VGAT deficient (VGAT^−/−^) mice were generated to elicit VGAT’s functional role and importance throughout development [1]. Unlike GAD67 deficient (GAD67^−/−^) mice that die within a few hours after birth due to respiratory failure, VGAT^−/−^ mice die between embryonic day (E) 18.5 and birth with a hunched posture [1,2]. In fact, VGAT^−/−^ mice displayed a more severe cleft palate and omphalocele phenotype than GAD67^−/−^ mice. Moreover, the lack of mobility and stiffness noted in VGAT^−/−^ mice in the initial study was later confirmed [186]. In the latter study, E18.5 VGAT^−/−^ foetuses were maintained alive in phosphate buffered saline following a caesarean section and were found suffering from severe impaired motor function as they failed to respond to mechanical stimuli (pinches to the tail) [186]. Significantly higher levels of GABA and glycine were found in these VGAT^−/−^ mice, presumably due to reduced uptake/degradation of these transmitters by neurons and glia as GAD67 and GAD65 expression was unchanged [186]. Furthermore, electrophysiological recordings from VGAT^−/−^ spinal cord motor neurons, hypoglossal motor neurons, and cultured neurons showed minimal to no spontaneous GABAergic or glycinergic postsynaptic currents, though a similar level of glutamatergic postsynaptic currents were detected in both control and VGAT^−/−^ motor neurons [2,67,186]. Interestingly, increased neuronal growth, including dendritic length, complexity, and somatic and dendritic spiny processes, were seen in hypoglossal motor neurons from VGAT^−/−^ mice, despite the lack of glycinergic and GABAergic neurotransmission during early embryonic development [67]. Despite the temporarily increased motor neuronal growth, motor neuron loss resulted, which is suspected to be the result of an imbalance in synaptic inhibition–excitation. These observations suggest that an increase in glutamatergic postsynaptic currents is likely a compensatory mechanism for decreased GABAergic or glycinergic postsynaptic currents during neurodevelopment, and have highlighted the negative effect of E/I imbalance in neuronal survival and neurodevelopment [67]. They also suggests that the sensory pathway to motor neurons is formed independently from GABA- and glycine-mediated synaptic transmission, and that the significant impact of VGAT deficiency is not limited to inhibitory neurotransmission, but influences embryonic development [186]. Although a number of studies have revealed the morphological, physiological, and electrophysiological alterations under the impact of VGAT deficiency in VGAT^−/−^ mice, further studies could investigate the role of the reduced level GABAergic or glycinergic postsynaptic currents in the neurodevelopmental phenotype seen in this mouse model. Thus, VGAT haplodeficient (VGAT^+/−^) mice provide another way of manipulating GABAergic and glycinergic transmission.

## 5. Concluding Remarks and Future Perspectives

Modulation of neural activity by the GABAergic system is crucial for normal neurodevelopment and subsequent neural circuit function. Studies have revealed strong links between deregulated inhibitory signalling and the pathophysiology of common neurological diseases, including epilepsy, schizophrenia, autism spectrum disorders, and anxiety. Further research to examine alterations in neuromorphology, neurophysiology, and behaviour of the GAD67^+/−^ and VGAT^−/−^ mice is likely to provide invaluable insights into disorders of synaptic inhibition and their prevention and treatment.

## Figures and Tables

**Figure 1 ijms-23-07965-f001:**
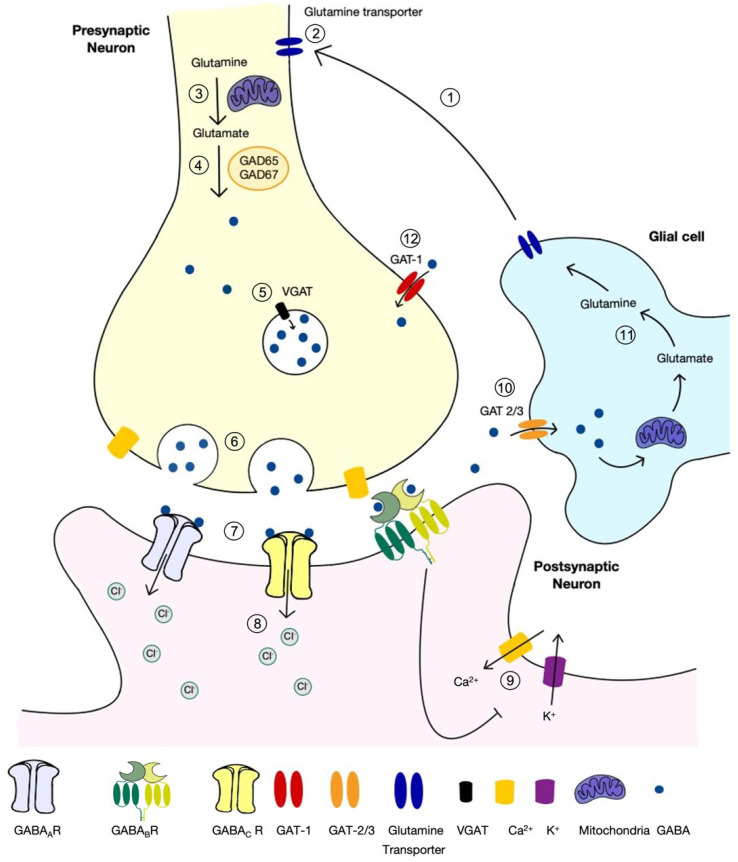
The mechanism of GABAergic neurotransmission. GABA is synthesised in the cytoplasm of the presynaptic terminal by GAD65 or GAD67 from glutamate supplied by adjacent glial cells via uptake of glutamine by the glutamine transporter and conversion to glutamate (1 to 4). GABA is then loaded into synaptic vesicles by VGAT (5) and released into the synaptic cleft via vesicle exocytosis (6). GABA can then bind to ionotropic GABA_A_ or GABA_C_ receptors (7), directly activating Cl^−^ flux across the postsynaptic membrane (8), or to metabotropic GABA_B_ receptors, indirectly activating Ca^2+^ or K^+^ ion channels via second messenger signalling (9). GABA is taken up from the synaptic cleft into glial cells via GAT 2/3 (10) for breakdown into glutamine (11) or into the presynaptic terminal via GAT-1 for recycling into synaptic vesicles (12).

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
