# Peer review of "Investigating the Role of GABA in Neural Development and Disease Using Mice Lacking GAD67 or VGAT Genes"

_ijms, 2022, doi:10.3390/ijms23147965_

Round 1

Reviewer 1 Report

Bolneo et all provide a nicely written, well organized and comprehensive review of GABAergic signaling during development in the mammalian brain. I offer a few suggestions for improvement:

1 - “system involves balance between excitatory” should be “system involves a balance between excitatory”

2 – “Neurological disorders that arise from deregulated synaptic inhibition have a devastating impact on the livelihoods of many around the world.” Sentence should be deleted since it is out of place and adds nothing to the intro paragraph.

3 – “synaptic inhibition will lead to a positive” should be “synaptic inhibition may lead to a positive”

4 – “a variety of neurodevelopmental and neuropsychiatric disorder” should be “a variety of neurodevelopmental and neuropsychiatric disorders”

5 – “known as seizures, which is observed in” should be “known as seizures, which are observed in”

6 – “Evidently, the hippocampus and cerebral cortex are most prone to epileptogenesis, due to their lower seizure threshold” should be “The hippocampal and cerebral cortices are most prone to epileptogenesis, as they exhibit the lowest seizure thresholds”

7 – “or by promoting neurotransmitter or voltage-gated” should be “or by enhancing neurotransmitter or voltage-gated”

8 – “FIGURE 1” Line 530 should be deleted.

I congratulate the Authors for a timely and informative review of the GABA-developmental story.

Author Response

Academic Editor Comments

Point-by-point response

Manuscript number: ijms-1726782

Corresponding Author: Peter G Noakes

Reviewing Editor (RE)

RE-pt 1: The title should be more specific and consistent with the content of the manuscript.

Response - We have altered the title to “Investigating the role of GABA in neural development and disease using mice lacking GAD67 or VGAT genes” (lines 1-2).

RE-pt 2: The authors propose to discuss GAD67 and VGAT mutant mice, and the manuscript should focus more on the systematic review of these animal models instead of promising to discuss the extremely broad topic of "The role of GABA in neural development and disease", which is only very superficially and incompletely covered in the submitted manuscript.

Response – We have responded to this opening comment, by addressing point by point the comments and suggestions of Reviewer #1 and Reviewer #2. All comments are addressed. We have indicated by line number, where specific improvements to the manuscript are located. Further, new text is in blue to aid our Reviewers and Editor in their assessment of our revised manuscript.

Reviewer#1 (R1) Round 1 (Comments and Suggestions for Authors)

Bolneo et all provide a nicely written, well organized and comprehensive review of GABAergic signalling during development in the mammalian brain. I offer a few suggestions for improvement: 

R1-pt1 - “system involves balance between excitatory” should be “system involves a balance between excitatory” 

Response: Changes has been made according to reviewer’s suggestion in Line 17-18.

R1-pt2 – “Neurological disorders that arise from deregulated synaptic inhibition have a devastating impact on the livelihoods of many around the world.” Sentence should be deleted since it is out of place and adds nothing to the intro paragraph.                                             

Response: This sentence has been deleted, according to reviewer’s suggestion, in paragraph 1, page 3.

R1-pt3 – “synaptic inhibition will lead to a positive” should be “synaptic inhibition may lead to a positive”                                                                                                                                Response: Changes has been made according to reviewer’s suggestion in Line 44.

R1-pt4 – “a variety of neurodevelopmental and neuropsychiatric disorder” should be “a variety of neurodevelopmental and neuropsychiatric disorders”                                                                    Response: Changes has been made according to reviewer’s suggestion in Line 198-199.

R1-pt5 – “known as seizures, which is observed in” should be “known as seizures, which are observed in” 

Response: Changes has been made according to reviewer’s suggestion in Line 215-216.

R1-pt6 – “Evidently, the hippocampus and cerebral cortex are most prone to epileptogenesis, due to their lower seizure threshold” should be “The hippocampal and cerebral cortices are most prone to epileptogenesis, as they exhibit the lowest seizure thresholds”                   

Response: Changes has been made according to reviewer’s suggestion in Line 235-236.

R1-pt7 – “or by promoting neurotransmitter or voltage-gated” should be “or by enhancing neurotransmitter or voltage-gated”                                                                                       Response: Changes has been made according to reviewer’s suggestion in Line 258-259.

R1-pt8 – “FIGURE 1” Line 530 should be deleted.                                                            Response: “FIGURE 1”  in line 512 of original text  is deleted (line above Figure 1).

I congratulate the Authors for a timely and informative review of the GABA-developmental story. Response: We thank our reviewer for their supportive comment.

Reviewer 2 Report

In their manuscript entitled “The role of GABA in neural development and disease” Erika Bolneo and coworkers present a review article that ought to summarize the “the involvement of down-regulated inhibitory signalling in neurological disorders, possible mechanisms for disease progression, and targets for therapeutic intervention” (line 17) and aim “to highlight how useful the GAD67+/- 34 and VGAT-/- mouse models are for the neuroscientific community” (Line 34). While the general topic of the manuscript, to entangle the role of inhibitory neurotransmitter dysfunction in the etiology of neurodevelopmental, is a valid question (although already covered by several recent review articles), in my opinion the present manuscript does not provide this information in a way that can be useful for most readers. Therefor I suggest that the authors resubmit this manuscript after a major revision. Please find several of my concerns in the following:

1.) Incomplete description of GABAergic effects during development. (Line 147, see also line 166-167). There are several articles available (e.g. by Ben-Ari, Cherubini, Fukuda, Holthoff, Khazipov, or Luhmann), that reviewed in detail the contribution of GABAergic processes on neuronal development. In order to keep your manuscript valuable and to convince the reader that GBA dysfunction contributes to the etiology of neuropsychiatric diseases and that the GAD-/- mouse is an adequate model, you should provide all of the relevant data in a concise but precise manner. In contrast to your statement that “GABA has been demonstrated to provide significant excitatory activity in the developing CNS” recent results demonstrate that the functional implications of depolarizing GABA responses are more complicated that the simple depolarizing = excitatory scheme proposed by Ben-Ari. Here the authors should definitely refer to the recent studies on this topic (e.g. Kaila group, Kirmse & Holthoff, Khazipov group, Colonesse and coworkers). In addition, in the light of their topic the authors should also refer more to the contribution of GABAergic transmission to the development of early network patterns, which are crucial for the establishment of neuronal connectivity. Please also note that the observation that the “…shift from an excitatory to an inhibitory action of GABA is modulated in an activity-dependent manner...” (Ganguly et al.) is highly controversial.

2.) You may consider also to provide a section about the role of GABA-B receptors (or otherwise state that you exclude the effect of GABA-B receptors). A statement like “GABA is the primary inhibitory neurotransmitter, binding to GABA-A receptors, opening chloride channels and hyperpolarizing the cell” (Line 16) is clearly to simplifying, even for the abstract. In this respect also the description of GABA-B receptor activation is incomplete. While it is honorable to include recent molecular details about the GABA-B receptor activation sequences it may be equally relevant to show how the downstream electrical responses are mediated. These mechanisms are probably more relevant to explain the slower responses of GABA-B receptors (Line 99-104).

3.) In general, the structure of the review appears unordered. E.G

- you mention several times the topic of the review (e.g. in line 43 and line 83),

- in the section about the GABAergic function in the mature CNS you did not mention vGAT or GAT1/2, although these mechanisms are shown in Fig. 1 and are required for later sections of the manuscript (where they are then explained).

- you describe effects of mutation in the GADs in detail before outlining GABA receptors and other elements of GABAergic transmission (line 85),

- in the VGAT para (Line 124-130) mechanisms and observations are listed without providing a clear structure (e.g. mode of action, affinities, expression),

- the subchapter “GABAergic and glycinergic signalling during neurodevelopment” (Line 132-143) provides only a scetchy version of the information provided in the next subchapter (“GABA in Neurodevelopment”),

- In the schizophrenia chapter you mention a lot of mutations related to the GABAergic system (GAT-1, NL2) which would have better been described in an initial chapter summarizing all elements of GABAergic transmission and synaptogenesis,

- in Lines 382-389 you repeat the information the sequence of neurodevelopmental events described in detail before.

4.) Restriction to GABAergic effects. While from the title the reader assume that the focus of the review article is on the GABAergic systems, you mention at many instances glycinergic transmission. While I, of course, understand that for this topic you should also shortly mention the function of glycine receptors, within the manuscript you do not carefully discriminate between both effects. In my opinion the mentioned glycinergic phenotypes (Line 198-201) do not contribute much to the main topic of the review. Also for most parts you concentrate on GABAergic processes and thus the parallel mentioning of glycinergic receptors in several sentences is more distracting than supportive. Therefore I would suggest to concentrate only on GABA.

5.) “GABA is released at inhibitory synapses …” (Line 47). In my opinion you should also mention here the non-synaptic release of GABA (and other endogenous GABAergic transmitters) that most probably is highly relevant for early developmental processes.

6.) Improper references. E.G.

-Chalin & Saha (2010) is probably not the best reference to support the statement “A growing focus of research is dedicated towards revealing the critical role of inhibitory neurotransmitters in refining many aspects of neurodevelopment”,

- Kirsch (2006) is in my opinion not the best reference that a balanced neuronal activity is required for neuronal development (there a several excellent reviews on this topic available),

-there are much more distinctive and general descriptions than Fogaty et al 2016/2013 available to support the statement that “there is a strong relationship between altered synaptic activity onto neurons and concomitant neuronal dendritic architecture and spine density changes”.

- There are more recent review article of the properties of cation-chloride cotransporters than Payne 2003 (e.g. from Blaesse et al, of Fukuda). In particular, also consider the critical recent review by Kaila & Hübner.

- Behar et al. (2001) is restricted to effects of GABA-B Receptors on migration. In lines 155-156 you should also cite references demonstrating the contribution of GABA-A and GABA-C receptors.

7.) Relevance of the VGAT-/- mice as model for neurodevelopmental disorders: In my opinion the VGAT-/- mouse, or at least the results you mention within the manuscript, does not profoundly contribute to our understanding of the etiology of neurodevelopmental diseases like schizophrenia of ASD. I think to most readers it is not surprising that a complete lack of synaptic GABA release will lead to massive alterations in the neuronal development (given the GABAergic effects from neurogenesis to synptogenesis). Are there studies on VGAT+/- mice available, which may allow to estimate the effects of more subtle changes in GABAergic function. On the other hand, you may use this model to speculate about the reason why a lack of synaptic GABA/glycine release has so little effect on prenatal neuronal development, when discussing the contribution of GABAergic activity to vyrious steps of neuronal development.

Minor:

1.) Line 39: “will lead to a positive change in…” Please tone down this statement.

2.) Line 69. Please rephrase, either “GAD65- and GAD67-mediated synthesis of GABA” or “GABA synthesis by GAD65 and GAD 67..”.

3.) Line 93: Benzodiazepines, barbiturate and picrotoxin cannot be termed “inhibitory molecules”. Probably they can be summarized as “modulatory molecules”.

4.) Line 94. “Variations within the strength and rate of signal elicited by GABA receptors are dependent upon whether they are an ion channel (GABA-A receptors) or coupled with a G-protein (GABA-B receptors)”. This statement neglects other factors that contribute to the properties of GABAergic signaling, like (GABA(A) receptor subunits, plasticity etc. Consider rephrasing.

5.): Line 104: “Moreover, the conjugation of different subunit isoform combinations can lead to a variety of physiological profiles for inhibitory neurotransmission, allowing for enhanced inhibitory control within the CNS”. To my knowledge one should better speak of “splice variants” than of subunit variants, as only 2 GABA-B subunits for the heterodimeric GABA-B receptors has been described. In addition, I missed a statement like this for the GABA-A receptors, where the subunit diversity profoundly determine the properties of GABAergic responses.

6.) Line 149: You may consider to include here also the role of GABA for the proliferations and migration of GABAergic interneurons (e.g. Haydar et al, Bartone & Polleux).

7.) Line 171: Here, in the light of the review topic, the author should discriminate between an “altered development of the GABAergic system” (i.e. the different elements of the inhibitory modulation provided by the various GABAergic interneuron subtypes) and an altered development of brain circuits caused by improper GABA function during development.

8.) Line 203: Omit the phrase “and social behaviour” or rephrase the sentence.

9.) Line 300: Change reference style for (Roberts, 1972).

10.) Line 315: “Hence, reduction of GAT-1 would lead to prolonged accumulation of GABA at the synaptic cleft and increased GABA-mediated inhibitory synaptic transmission”. However, in the sentence before you mention that GAD65/67 and GABA receptor subunits are downregulated, which will drastically reduce the inhibitory impact. So the GAT1 decrease in the only effect bolstering GABA action and might be a compensatory effect. Please rewrite to emphasize the contradictory effects.

11.) Line 382: I would not follow the statement that the schedule of neurodevelopmental events is “rigid”, in fact it is rather plastic.

12.) In the respect of the GAD67+/- mouse and neurodevelopment, I may also be worth mentioning what has been learned from the GAD67-/- mouse for neuronal proliferation and migration.

13.) Line 432 Consider rephrasing “deficits in …. synapse formation on pyramidal neurons”.

14.) Line 503. Please correct the typo “Furthermore”

Author Response

Reviewer#2 (R2) Round 1 (Major concerns)

In their manuscript entitled “The role of GABA in neural development and disease” Erika Bolneo and coworkers present a review article that ought to summarize the “the involvement of down-regulated inhibitory signalling in neurological disorders, possible mechanisms for disease progression, and targets for therapeutic intervention” (line 17) and aim “to highlight how useful the GAD67+/- 34 and VGAT-/- mouse models are for the neuroscientific community” (Line 34). While the general topic of the manuscript, to entangle the role of inhibitory neurotransmitter dysfunction in the etiology of neurodevelopmental, is a valid question (although already covered by several recent review articles), in my opinion the present manuscript does not provide this information in a way that can be useful for most readers. Therefor I suggest that the authors resubmit this manuscript after a major revision. Please find several of my concerns in the following:

R2 pt 1: Incomplete description of GABAergic effects during development. (Line 147, see also line 166-167). There are several articles available (e.g. by Ben-Ari, Cherubini, Fukuda, Holthoff, Khazipov, or Luhmann), that reviewed in detail the contribution of GABAergic processes on neuronal development.  In order to keep your manuscript valuable and to convince the reader that GBA dysfunction contributes to the etiology of neuropsychiatric diseases and that the GAD-/- mouse is an adequate model, you should provide all of the relevant data in a concise but precise manner. In contrast to your statement that “GABA has been demonstrated to provide significant excitatory activity in the developing CNS” recent results demonstrate that the functional implications of depolarizing GABA responses are more complicated that the simple depolarizing = excitatory scheme proposed by Ben-Ari. Here the authors should definitely refer to the recent studies on this topic (e.g. Kaila group, Kirmse & Holthoff, Khazipov group, Colonesse and coworkers). In addition, in the light of their topic the authors should also refer more to the contribution of GABAergic transmission to the development of early network patterns, which are crucial for the establishment of neuronal connectivity. Please also note that the observation that the “…shift from an excitatory to an inhibitory action of GABA is modulated in an activity-dependent manner...” (Ganguly et al.) is highly controversial.                                                                                                                                  

Response: We thank our review for these comments and suggestions. As suggested, we have now referred to the above-mentioned reviews of developmental actions of GABA. We have amended line 140-143 to read “While GABA is the major inhibitory neurotransmitter in the mature mammalian CNS, GABA-A receptors evoke excitation before birth and this period of excitation can even extend into early postnatal life, evoking depolarisation in postsynaptic neurons [46,49-54].” [46 = Virtanen, M.A]; [49 = Ben-Ari, Y.,]; [50 = Schmidt, M.J. and K. Mirnics]; [51 = Murata, Y. and M.T. Colonnese]; [53 = Kirmse, K., et al.]; and [54 = Watanabe, M. and A. Fukuda].”  We have also amended lines 157-158 to read “The shift from an excitatory to an inhibitory action of GABA may also be modulated in an activity-dependent manner [49].)

R2-pt 2: You may consider also to provide a section about the role of GABA-B receptors (or otherwise state that you exclude the effect of GABA-B receptors). A statement like “GABA is the primary inhibitory neurotransmitter, binding to GABA-A receptors, opening chloride channels and hyperpolarizing the cell” (Line 16) is clearly to simplifying, even for the abstract. In this respect also the description of GABA-B receptor activation is incomplete. While it is honorable to include recent molecular details about the GABA-B receptor activation sequences it may be equally relevant to show how the downstream electrical responses are mediated. These mechanisms are probably more relevant to explain the slower responses of GABA-B receptors (Line 99-104).                                                                                                                

Response: We thank our reviewer for these suggestions and have deleted the over simplified statement from original line 16 (Abstract) of our manuscript. We have also amended line 76-81 exclude GABA-B and GABA-C receptors from consideration in the review- as suggested.  This edited text now reads “The effects of GABA are elicited by the binding of GABA to the ionotropic GABA-A and GABA-C receptor subtypes and the metabotropic GABA-B receptor subtype on the postsynaptic membrane of neurons [24 (Kelsom, C. and W. Lu,), 25 (Sigel, E. and M.E. Steinmann,)] (Figure 1). This review will focus on the role of GABA-A receptors, which form the majority of GABA receptors within the CNS are GABA-A receptors, which incorporate an ion channel that allows the passage of chloride anions, and exert primary control over inhibitory signaling [26 (Olsen, R.W., DeLorey, T. M)].

R2-pt3: In general, the structure of the review appears unordered. E  .G                             Response: we have reordered sections of the review and edited text to improve structure.  The revised subheadings are:

PREFACE (line 33).

IMPORTANCE OF GABA IN THE NERVOUS SYSTEM (line 45) Inhibition in the forebrain is mediated by GABA; The role of GABA in the mature central nervous system; The role of VGAT in GABA and glycine signalling; GABA in neurodevelopment; Excitatory- inhibitory balance is crucial in normal neurodevelopment.

DISORDERS OF SYNAPTIC INHIBITION Synaptic inhibition dysfunction in epilepsy; Synaptic inhibition dysfunction in schizophrenia: Autism Spectrum Disorders and GABAergic dysfunction;

GABA DEFICIENT MOUSE MODELS: The GAD67+/- mouse; The VGAT deficient mouse;

CONCLUDING REMARKS AND FUTURE PERSPECTIVE

R2-pt4- you mention several times the topic of the review (e.g. in line 43 and line 83),.               

Response:  original line number 43 “Our aim is to highlight how useful the GAD67+/- and VGAT-/- mouse models are for the neuroscientific community.” has been deleted (original line 43). “After demonstrating the vital role GABA plays in central nervous system (CNS) inhibition, we must briefly outline its mechanisms of action, synthesis, and the effects of mutation in GABA-producing enzymes.” has been deleted  (original line 78).

R2-pt5 - in the section about the GABAergic function in the mature CNS you did not mention vGAT or GAT1/2, although these mechanisms are shown in Fig. 1 and are required for later sections of the manuscript (where they are then explained).                                                      

Response: We have clarified the role of GABA transporters GAT-1 and GAT2/3 and cited Figure 1 (Lines 88-94). The section on VGAT has been moved forward to immediately follow the role of GABA in the mature nervous system, lines 117-121.

R2-pt6 - you describe effects of mutation in the GADs in detail before outlining GABA receptors and other elements of GABAergic transmission (line 85).                                            

Response: description of the effects of mutating GAD isoforms has been moved to lines 305-309 (after description of GABA receptors).

R2-pt7- in the VGAT para (Line 124-130) mechanisms and observations are listed without providing a clear structure (e.g. mode of action, affinities, expression).                            

Response: We agree that the original structure was unclear and have rearranged the structure of the VGAT paragraph in the following order, mode of action, affinities, and expression (new lines 115-119.

R2-pt8 - the subchapter “GABAergic and glycinergic signalling during neurodevelopment” (Line 132-143) provides only a sketchy version of the information provided in the next subchapter (“GABA in Neurodevelopment”),                                                                 

Response: This subchapter section has been deleted. (at original line 125). We have also modified the sub-heading to “GABA in neurodevelopment” (line122).

R2-pt9 - In the schizophrenia chapter you mention a lot of mutations related to the GABAergic system (GAT-1, NL2) which would have better been described in an initial chapter summarizing all elements of GABAergic transmission and synaptogenesis.                                  

Response: A brief introduction of both GAT-1 and NL2 have been added into the subsection “The role of GABA in the mature central nervous system” in Lines 88-94.

R2-pt10 - in Lines 382-389 you repeat the information the sequence of neurodevelopmental events described in detail before.                                                                                     

Response: We have amended this to read “Additionally, ASD core behavioural symptoms are associated with the comorbid neurological disorders discussed above, such as epilepsy and schizophrenia, as well as other conditions associated with dysfunctional GABAergic neurotransmission, such as Parkinson’s disease (PD), bipolar disorder, anxiety and depression [35 (Gaetz, W., et al.,), 71 (Moy, S.S., et al.,), 155 (Coghlan, S., et al.)].” Lines 386-389.

R2-pt11 - Restriction to GABAergic effects. While from the title the reader assume that the focus of the review article is on the GABAergic systems, you mention at many instances glycinergic transmission. While I, of course, understand that for this topic you should also shortly mention the function of glycine receptors, within the manuscript you do not carefully discriminate between both effects. In my opinion the mentioned glycinergic phenotypes (Line 198-201) do not contribute much to the main topic of the review. Also for most parts you concentrate on GABAergic processes and thus the parallel mentioning of glycinergic receptors in several sentences is more distracting than supportive. Therefore I would suggest to concentrate only on GABA.                                                                                             

Response: We have deleted sections on glycinergic neurotransmission and focussed on GABAergic effects

R2-pt12 - “GABA is released at inhibitory synapses …” (Line 47). In my opinion you should also mention here the non-synaptic release of GABA (and other endogenous GABAergic transmitters) that most probably is highly relevant for early developmental processes.                    

Response: We have included the following “In addition, since many events of neurogenesis occur before the onset of synapse formation, non-synaptic GABAergic transmission together with endogenously released GABA has been postulated to be involved in early neurodevelopment [6 (Kilb, W., S. Kirischuk, and H.J. Luhmann,)]. Tonic GABA release [7 (Cellot, G. and E. Cherubini,), 8  (Manent, J.-B., et al.)] lays down the foundation for normal proliferation of neuronal precursors, neuronal differentiation and migration, and early activity patterns [6 (Kilb, W., S. Kirischuk, and H.J. Luhmann)].” Lines 53-59.

R2-pt13 - Improper references. E.G.-Chalin & Saha (2010) is probably not the best reference to support the statement “A growing focus of research is dedicated towards revealing the critical role of inhibitory neurotransmitters in refining many aspects of neurodevelopment”

Response: this reference has been deleted and replaced by references – [7, (Cellot, G. and E. Cherubini)],[8, (Manent, J.-B., et al.)]) and [42,( Bruining, H., et al.)] (line 124).

R2-pt14 - Kirsch (2006) is in my opinion not the best reference that a balanced neuronal activity is required for neuronal development (there a several excellent reviews on this topic available).                                                                                                                            

Response: this reference has been deleted and replaced by references [42, (Bruining, H., et al. 2020)] and [47,( Gatto, C.L. and K. Broadie, 2010)](Line 138 )].

R2-pt15 - there are much more distinctive and general descriptions than Fogaty et al 2016/2013 available to support the statement that “there is a strong relationship between altered synaptic activity onto neurons and concomitant neuronal dendritic architecture and spine density changes”.                                                                                                                                   

Response:  We have included these references as they are good examples of single cell correlation of electrophysiological responses and neuronal morphology in the absence of GABA/Glycine neurotransmission (references 13, 67 and 70]. We have as also as suggested, included three additional references (references, 66, 68, and 69) - Lines 177-183. [13 = Fogarty, M.J., et al.,2016]; [66 = Villa, K.L. and E. Nedivi,]; [67 = Fogarty, M.J., et al. 2017] [68 = Forrest et al]; [69 = Penzes, P., K.M. Woolfrey, and D.P. Srivastava], [70 = Klenowski, P.M., et al.] 

R2-pt16 - There are more recent review article of the properties of cation-chloride cotransporters than Payne 2003 (e.g. from Blaesse et al, of Fukuda). In particular, also consider the critical recent review by Kaila & Hübner.                                                           

Response: We thank our reviewer, and have added two new references [45 = (Blaesse et al (2009)], and [46 = Virtanen, Uvarov, Hubner and Kaila ], line 129.

R2-pt17 -  Behar et al. (2001) is restricted to effects of GABA-B Receptors on migration. In lines 155-156 you should also cite references demonstrating the contribution of GABA-A and GABA-C receptors.                                                                                                                     

Response: We have deleted “ GABA-B and GABA-C receptors are then responsible for maintaining cell migration, while … (original line 149)”, including the Behar et al., 2001 citation. We have made it clear that the properties of cell migration are restricted to GABA and have appropriate citations [57 = Haydar, T.F., et al.]; [58 = Bortone, D. and F. Polleux]; [59 = LoTurco, J.J., et al.]; and [60 = Wang, Y., et al.], lines 147 -150.

R2-pt18 - Relevance of the VGAT-/- mice as model for neurodevelopmental disorders: In my opinion the VGAT-/- mouse, or at least the results you mention within the manuscript, does not profoundly contribute to our understanding of the etiology of neurodevelopmental diseases like schizophrenia of ASD. I think to most readers it is not surprising that a complete lack of synaptic GABA release will lead to massive alterations in the neuronal development (given the GABAergic effects from neurogenesis to synptogenesis). Are there studies on VGAT+/- mice available, which may allow to estimate the effects of more subtle changes in GABAergic function. On the other hand, you may use this model to speculate about the reason why a lack of synaptic GABA/glycine release has so little effect on prenatal neuronal development, when discussing the contribution of GABAergic activity to various steps of neuronal development.       

Response: We thank our reviewer for pointing out that VGAT-/- mouse is not the most relevant model for understanding neurodevelopmental diseases like schizophrenia of ASD, which we agree. However, it is a model that is deficient in the major vesicular transporter of GABA, which was expected to have a detrimental effect on neurodevelopment and neuronal growth. It is indeed not surprising that the lack of VGAT could cause massive alterations in the neuronal development (given the GABAergic effects from neurogenesis to synaptogenesis), but increased neuronal growth, including dendritic length, complexity, and somatic and dendritic spiny processes, were seen in VGAT-/- mice ([67 = Fogarty et al 2017, line 495). These observations confirmed and highlighted the significance of excitatory and inhibitory imbalance, which is an emerging common phenomenon in different neurodevelopmental, psychiatric and neurodegenerative disorders as we have mentioned in this review.                              We have added more details regarding the initial increased neuronal growth and eventual motor neuron loss in the paragraph “Despite the temporarily increased motor neuronal growth, MNs loss was resulted essentially, which is suspected to be the result of imbalance in synaptic inhibition in excitation. These observations suggest that an increase in glutamatergic postsynaptic currents is likely a compensatory mechanism for decreased GABAergic or glycinergic postsynaptic currents during neurodevelopment, and have highlighted the negative effect of E/I imbalance in neuronal survival and neurodevelopment [67]”, Lines 495-500.  We do acknowledge that further studies could investigate what is the role of the reduced level GABAergic or glycinergic postsynaptic currents in the neurodevelopmental phenotype seen in the VGAT -/- mouse model. Thus, VGAT haplodeficient (VGAT+/-) mice provide another way of manipulating GABAergic and glycinergic transmission.”  These details have been added – lines 500-509.[186 = Saito, K., et al.,]

R2-Minor: 

R2-minor-1 - Line 39: “will lead to a positive change in…” Please tone down this statement.      Response: “will lead to a positive change in…” is changed into “may also lead to a positive change in…” in Line 44.

R2-minor-2 - Line 69: Please rephrase, either “GAD65- and GAD67-mediated synthesis of GABA” or “GABA synthesis by GAD65 and GAD 67.”                                                     Response: We have amended this to “GAD67- and GAD65-mediated synthesis of GABA differ in a temporospatial manner [30].” Line 98-99

R2-minor-3 - Line 93: Benzodiazepines, barbiturate and picrotoxin cannot be termed “inhibitory molecules”. Probably they can be summarized as “modulatory molecules”.        Response: We have changed the term inhibitory molecules to “modulatory molecules” in Line 85.

R2-minor-4 - Line 94: “Variations within the strength and rate of signal elicited by GABA receptors are dependent upon whether they are an ion channel (GABA-A receptors) or coupled with a G-protein (GABA-B receptors)”. This statement neglects other factors that contribute to the properties of GABAergic signaling, like (GABA(A) receptor subunits, plasticity etc. Consider rephrasing.                                                                                              Response: This statement has been deleted at line 112

R2-minor-5 - Line 104: “Moreover, the conjugation of different subunit isoform combinations can lead to a variety of physiological profiles for inhibitory neurotransmission, allowing for enhanced inhibitory control within the CNS”. To my knowledge one should better speak of “splice variants” than of subunit variants, as only 2 GABA-B subunits for the heterodimeric GABA-B receptors has been described. In addition, I missed a statement like this for the GABA-A receptors, where the subunit diversity profoundly determine the properties of GABAergic responses.                                                                                                         Response: This statement has been deleted at line 112

R2-minor-6 - Line 149: You may consider to include here also the role of GABA for the proliferations and migration of GABAergic interneurons (e.g. Haydar et al, Bartone & Polleux).                                                                                                                                  Response: We have added in the suggested references and ammened the text accordingly (lines 147-148 . [57 = Haydar, T.F., et al.,]; [58 = Bortone, D. and F. Polleux]; and  [59 = LoTurco, J.J., et al.,]

R2-minor-7 - Line 171: Here, in the light of the review topic, the author should discriminate between an “altered development of the GABAergic system” (i.e. the different elements of the inhibitory modulation provided by the various GABAergic interneuron subtypes) and an altered development of brain circuits caused by improper GABA function during development.                                                                                                                            Response We have as suggested, discriminated between alerted development of the GABAergic system and the altered output function of neural circuits due to changes in GABAergic synaptic transmission (lines 165-168).  

R2-minor-8 - Line 203: Omit the phrase “and social behaviour” or rephrase the sentence.   Response: “and social behaviour” has been removed from line 195.

R2-minor-9 Line 300: Change reference style for (Roberts, 1972).                                     Response: this citation has been deleted at line 290.

R2-minor-10 - Line 315: “Hence, reduction of GAT-1 would lead to prolonged accumulation of GABA at the synaptic cleft and increased GABA-mediated inhibitory synaptic transmission”. However, in the sentence before you mention that GAD65/67 and GABA receptor subunits are downregulated, which will drastically reduce the inhibitory impact. So the GAT1 decrease in the only effect bolstering GABA action and might be a compensatory effect. Please rewrite to emphasize the contradictory effects.           

 Response: We thank our reviewer for pointing out the reduction in GAT-1 could be a compensation mechanism or has a compensatory effect. The reduction of those essential GABAergic synaptic transmission components are observations various studies have made through human and animal studies. We have highlighted that the potential compensatory effect of the overall downregulation between GAD67, GAD65, and GAT-1 in Line 306-309.

R2-minor-11 - Line 382: I would not follow the statement that the schedule of neurodevelopmental events is “rigid”, in fact it is rather plastic.                                                Response: We have amended this section to make it clear that the schedule of neurodevelopment is plastic (see Lines 367-374).

R2-minor-12 - In the respect of the GAD67+/- mouse and neurodevelopment, it may also be worth mentioning what has been learned from the GAD67-/- mouse for neuronal proliferation and migration.                                                                                                              Response: There is a lack of evidence in the literature on the neuronal proliferation and migration on GAD67-/- mouse, but we agree that it is an interesting area to be investigated.

R2-minor-13  Line 432 Consider rephrasing “deficits in …. synapse formation on pyramidal neurons”.                                                                                                                          Response: The sentence is rephrased to “deficits in axon branching and abnormal pyramidal neuron synaptic formation were also observed in adult mouse” in Lines 418-419.

R2-minor-14 - Line 503. Please correct the typo “Furthermore”.                                        Response:  this has been done

Round 2

Reviewer 2 Report

Please note a typo in line 307, probably this should read  .... is a general downregulation of both GAD65 mRNA and GAD67 mRNA, .....".

However, this error can also be addressed in the proofs.